

# Comparative analysis of trade-offs and synergies in ecosystem services between Guanzhong Basin and Hanzhong Basin in China

Bo-Yan Li[1], Wei Wang[1,2], Yun-Chen Wang[3]

[1]State Key Laboratory of Information Engineering in Surveying, Mapping and Remote Sensing(LIESMARS), Wuhan University, Wuhan 430079, China
[2]Collaborative Innovation Center of Geospatial Technology, Wuhan 430079, China
[3]Key Laboratory of Remote Sensing of Gansu Province, Northwest Institute of Eco-Environment and Resources, Chinese Academy of Sciences, Lanzhou, Gansu 730000, China

*Correspondence to*: Bo-Yan Li (liboyan3900139@163.com)

**Abstract.** An important feature of the relationships among ecosystem services (ES) is they have temporal and spatial patterns. The purpose of this research was to study the spatial and temporal characteristics of the synergies and trade-offs in ES in Guanzhong Basin and Hanzhoung Basin, as well as to compare the ES differences between the two basins. The spatio-temporal characteristics of the relationships among ES were analysed and compared from 1995-2014 for Hanzhong Basin, which has a
good ecological environment, and the economically developed Guanzhong Basin, using linear relationship between grain output and NDVI (LRGO & NDVI), the Carnegie–Ames–Stanford Approach (CASA), the integrated storage capacity method(ISCM), and the revised universal soil loss equation (RUSLE) model to simulate the four types of ES: food production (FP), net primary production (NPP), water retention (WR) and soil conservation (SC). The results of this study were as follows: (1) The trade-off relationships between FP and NPP in Guanzhong Basin and Hanzhong Basin were the most significant, and
the trade-off relationship between FP and NPP in Guanzhong Basin (R=-0.40, P<0.01) was stronger than that in Hanzhong Basin (R=-0.31, P<0.01); the synergistic relationships between NPP and WR, as well as between WR and SC in Hanzhong Basin were stronger than those in Guanzhong Basin, but the synergistic relationship between NPP and SC was weaker in Hanzhong Basin than in Guanzhong Basin. (2) The synergistic relationships between WR and NPP, as well as between WR and SC weakened in the two basins over 20 years, and the trade-offs and synergies in the Hanzhong Basin were more significant
than those in Guanzhong Basin. (3) The spatial synergies and trade-offs between FP and WR, as well as between WR and SC was widespread in the two basins. This study will help to deepen the understanding of the relationship among ES in different areas because of regional differences in temporal and spatial patterns and will help broaden the depth and breadth of trade-offs and synergies in ES to provide a case for practical ES management under local conditions.

## 1 Introduction

Ecosystem services (ES) refer to the environmental conditions and effects of the survival and development of human beings that are formed and maintained by ecosystems, where all benefits are directly or indirectly derived from the ecosystems





(Costanza et al., 2017; TEEB, 2010). ES are not only the processes of maintaining the cycle of life materials, maintaining biodiversity, regulating meteorological processes, improving and maintaining soil, purifying the environment and so on(Tian et al., 2016), they are also the resource and environment foundation for the existence and development of human society (Wang et al., 2017). Demands for services are increasing because of economic development, population increase and living standards

improvement, and the supply ability of ecosystems is often improved by ecosystems transformations, such as land reclamation, building dykes to reclaim land from lakes, and reclaiming banned slopes. Although the supply ability of an ecosystem is greatly improved by such transformations, and food production(FP) is naturally greatly improved by the supply of services, WR, SC and other regulatory capacities are significantly reduced(Li and Wang, 2018). The relationships between ES have shown dynamic changes that generally present three relationship forms: trade-offs, synergies, and no relationship (Li et al., 2017).

Trade-offs refer to the decrease in supply of certain types of ES because of the increased use of other types. Synergies (or co-benefits) refer to the enhancement of two or more ES at the same time (Austrheim et al., 2016; Grace et al., 2014; Li and Wang, 2018). No relationship means that to two or more types of ES do not appear to increase or decrease the situation. These dynamic changes threaten the safety and health of human beings as well as affect the ecological security of the region, the state, and even the world (Li and Wang, 2018; Li et al., 2017). Therefore, it is urgent and necessary to study the trade-offs and synergies

among ES to better manage the various services of ecosystems, to find balance between economic development and environmental protection, and to balance realistic interests and future development.

The study of ES began with a book edited by Daily (Daily, 1997) and an article published in Nature on the value of the world's ES (Costanza et al., 1997). The Millennium Ecosystem Assessment (MEA)(Millennium Ecosystem Assessment (MEA), 2005) and economics of ecosystems and biodiversity (TEEB) (TEEB, 2010) have greatly advanced the study of ES. A large number

of scholars have analysed the identified the relationships between synergies and trade-offs (Feng et al., 2017; Grace et al., 2014; Li and Wang, 2018; Tian, Y., Wang, S., Bai, X., 2016), their manifestations (Ma et al., 2004), and spatio-temporal scale (Chen et al., 2017; Sun and Li, 2017) as well as the driving mechanisms and scenario changes of ES (Cervelli et al., 2017; Dai et al., 2016; Delphin et al., 2016; Yang et al., 2016). Many research methods have emerged, including the scenario analysis method (i.e., the inversion of land use and its effects at small regional extents (Clue-s), soil and water assessment tool (SWAT),

and integrated valuation of ES and trade-offs model (InVEST). The ES under scenarios as well as contrasting spatial patterns and scale effects can be analysed based in its dynamic changes under different land use dynamics, agricultural management, and forest management practices. The scientific and rationality of scenario plans are often questioned because the simulation of class activities and management scenarios has great subjectivity and variability (Yang et al., 2016). The service liquidity analysis method (the Artificial Intelligence for Ecosystem Services (ARIES) model) is a comparative analysis of the spatial

pattern and scale characteristics of ecosystem service supply, demand and transmission paths based on the principle of network analysis technology; however, it is only used for the trade-offs of a few types of ES, such as FP and fishery breeding (Wendland et al., 2010). Statistical methods (e.g., correlation analysis, regression analysis, cluster analysis and redundancy analysis) are usually used to analyse changes in the number of ES, and have the advantages of simplicity, speed, and lack of strict requirements for data types, the discrete area of aggregate data or sample data analysis will cover the spatial heterogeneity



within a region (Su et al., 2012). The spatial analysis method is based on geographic information system (GIS) technology, which is usually used for comparative analysis of spatial patterns and scale effects of ES and helps to analyse the mechanism of trade-offs and synergies in ES. However, the choices of spatial and temporal scales as well as spatial data resolution are very important and may filter out significant detailed information (Grace et al., 2014). Therefore, the correlation analysis and

spatial analysis methods were used in this study to combine the advantages of the spatial statistics and spatial analysis methods to select the appropriate spatial resolution to compare ES spatial pattern changes. Moreover, adding a time series analysis, allowed us to analyse the historical process of ES changes with time and reveal their development and change.

Over the past 5 years, scholars have studied the trade-offs and synergies among ES (Austrheim et al., 2016; Dai et al., 2016; Feng et al., 2017; Grace et al., 2014; Li and Wang, 2018; Sun and Li, 2017; Tian et al., 2016; Wang et al., 2017; Yang et al.,

2016). However, some problems remain: (1) the temporal and spatial heterogeneity of trade-offs and synergies have not been given sufficient, and the research on continuous time series has also been insufficient(Renard et al., 2015); (2) the studies of trade-offs or synergies have mainly been based on quantitative analysis of statistical relations to reflect regional overall differences, and there has been a lack of spatial expression of temporal and spatial differences within the region; and (3) trade-offs and synergies have rarely been considered in the study of spatio-temporal contrastive analysis. Therefore, according to

the proposed framework (Fig. 1), we studied four main ES (FP, NPP, WR and SC). Taking the south side of the Hanzhong Basin (humid area), which has a good ecological environment, and the north side of the Guanzhong Basin (sub-humid area), which is the economically developed, as example basins on both sides of the dividing line between the north and south of China as the research area, we analysed the spatio-temporal characteristics of the trade-offs and synergies among ES in the two basins and compared their differences to better plan for further policy adjustment.

## 2 Materials and Methods

### 2.1 Study Site

Guanzhong Basin (Fig.1a) is located in the centre of Shaanxi Province, China, between 33°41′ N to 35°55′ N and 106°19′ E to 110°35′ E (Fig.1c), and covers an area of approximately 39064.5 square kilometres. The Loess Plateau and the Qinling Mountains are north and south of Guanzhong Basin, respectively. The basin starts from the west of Baoji City

and continues to the end of Tongguan County. The average terrain is approximately 400 meters above sea level. With its fertile soil and the most socio-economically developed areas in Shaanxi Province, Guanzhong Basin has gained fame as the "800-mile Grain Basin" and is one of the major grain, cotton and oil-bearing crop producing areas. The area experiences a typically warm and semi-humid continental monsoon climate in which the four seasons are distinct and rainfall is moderate. The mean annual temperature is approximately 13.3 ℃, and the mean annual precipitation is

approximately 507.7 mm. The region has five cities (Xi'an City, Baoji City, Xianyang City, Weinan City, and Tongchuan City). The main land use types are dominated by farmland.



Hanzhong Basin (Fig.1d) is located in the south of Shaanxi Province, China, between 32°28′ N to 33°40′ N and 105°30′ E to 108°18′ E (Fig.1c), and covers an area of approximately 7364.8 square kilometres. The Qinling Mountains and Daba Mountains are north and south of Guanzhong Basin, respectively. The basin starts from the west of Wuhou town and continues to the end of Yang County. The average terrain is approximately 500 meters above sea level. With the

good ecological environment in Shaanxi Province, Hanzhong Basin has gained fame as the "Chiangnan in Northwest China". The area experiences a typical warm and humid subtropical climate; the mean annual temperature is approximately 15 ℃, and the mean annual precipitation is approximately 800 mm. The region is contained within the city of Hanzhong. The main land uses are predominately woodland and grassland.

Guanzhong Basin and Hanzhong Basin are both famous valley basins in northwest China. They are located on both sides

of the dividing line between north and south (the Qinling-Huaihe River), respectively (Fig. 2c). This study considered the unification of natural data and socio-economic data and chose the administrative boundary of Guanzhong Basin and Hanzhong Basin as the research area.

### 2.2 Data Sources

All data used in the ES model are listed in Table 1. The LUCC data were interpreted from the Landsat-5 TM (Thematic

Mapper), Landsat-7 ETM+ (Enhanced Thematic Mapper) and Landsat-8 OLI (Operational Land Imager) based on the classification method of decision trees. A spatial resolution of 30 meters met the requirements of regional scale. In this study, the land use types were divided into six main categories: farmland, woodland, grassland, water body, built-up land and other. Additionally, most of the remote sensing data were distributed in "June" or "September" (Fig. 3), and some data were not present or subject to cloudiness (e.g., year:1997; strip number:126; row number:035). "June" refers to a similar month in June

or around June, and the same was true for "September". Considering the seasonal variation of partial features and quality of the remote sensing data, the remote sensing data from the previous year's December or subsequent year's January data were replaced. The decision tree was generated using eCognition Developer 8.7 (Trimble, 2011) and established according to the logical relation after the result of two LUCC classifications. Then, the classification results were compared with Google Earth Pro 7.1.5 to achieve precise verification. The overall accuracy of the LUCC was over 87.5%, and the Kappa coefficients were

above 0.82 for all 20 years (1995-2014); 20 years of raster LUCC data were finally obtained (Fig. 4). The interpretation process was as follows:

(1) Water body: the water body type is different from other land use types, and the results were the best in two periods. Therefore, water bodies were extracted first.

(2) Other: in "September", the other and farmland types were very different and the floodplain area was wider than that in

"June"; the other type could thus be extracted.

(3) Farmland: because of the confusion of farmland and some built-up land, if the land use type of "June" or "September" met the conditions of farmland, it was extracted as farmland.




(4) Woodland and grassland: Woodland and grassland are difficult to distinguish, and the NDVI data were thus introduced; based on the sampling point and visual interpretation, the NDVI threshold value to distinguish woodland and grassland was 0.53 (manual).

(5) Built-up land: if the land use types of "June" and "September" satisfied the conditions of built-up land, the land use was extracted as built-up land.

(6) Accuracy verification: Validation samples were obtained by random sampling and simultaneous use if the Google Earth pro 7.1.5 software on historical images for visual interpretation;each category sampled 60 samples for a total of 360 samples, and the confusion matrix was obtained by Kappa coefficients and overall accuracy.

**Table 1**

Data types and sources

| Data type | Data sources |
|---|---|
| Landsat-5 TM (1995-2011) | Data set provided by the Geospatial Data Cloud site, Computer Network Information Center, |
| Landsat-7 ETM+ (2012) | Chinese Academy of Sciences: http://www.gscloud.cn |
| Landsat-8 OLI (2013-2014) | United States Geological Survey (USGS): http://landsat.usgs.gov/ |
| Digital Elevation Model (DEM) | Data set provided by the Geospatial Data Cloud site, Computer Network Information Center, Chinese Academy of Sciences: http://www.gscloud.cn |
| Soil and vegetation type data | Data set provided by the Data Center for Resources and Environmental Sciences, Chinese Academy of Sciences (RESDC): http://www.resdc.cn |
| Meteorological data (e. g., temperature, precipitation, sunshine duration and solar radiation) | Climatic Data Center, National Meteorological Information Center, China Meteorological Administration: http://data.cma.cn |
| Social-economic data | "Statistical Yearbook of Shaanxi", "Statistical Yearbook of Xi'an", "Statistical Yearbook of Tongchuan", "Statistical Yearbook of Xianyang", "Statistical Yearbook of Baoji", "Statistical Yearbook of Weinan", and "Statistical Yearbook of Hanzhong": http://tongji.cnki.net/overseas/brief/result.aspx |
| Other data (e. g., road traffic, water systems, administrative boundaries) | Data set provided by the Data Center for Resources and Environmental Sciences, Chinese Academy of Sciences (RESDC): http://www.resdc.cn |
| LUCC data | LUCC data were interpreted from the Landsat-5 TM, Landsat-7 ETM+ and Landsat-8 OLI |

## 2.3. Quantification of ES

The four ES (i.e., FP, NPP, WR and SC) were quantified using model simulations, as shown in Table 2. The methods of assessing food production services were mainly divided into two categories. First, they were simulated and predicted by the ecosystem productivity model. Second, the actual food production and farmland areas were evaluated and forecasted based on the statistical yearbook. Past studies have generally used various types of food production (cereals, beans, potatoes) or stocks as indicators of ecosystem food production without taking into account other land types (e.g., water, grassland) to provide



production services(Yang et al., 2016). However, food sources for human beings are far greater than food production and include other crops that can provide the nutrients needed for human survival, such as oil, sugar, meat, milk, eggs, and aquatic products. Animal husbandry and aquaculture, as well as imported and exported food, provide important inputs to total food production. To comprehensively assess the food production capacity of the study area and analyse the temporal and spatial

analysis of trade-offs and synergies with other ES, this study combined the LUCC types data and statistical yearbook data to simulate the total food output value of the LUCC types in the research area. The linear relationship between food production and NDVI was used to achieve the spatialization.

The many computational models of net primary production (NPP) are mainly divided into three categories: statistical models, parameter models and process models. The carnegie-ames-stanford approach (CASA) model is a process model based on

remote sensing light energy utilization. Considering the differences between biological parameters (e.g., NPP and vegetation coverage), the data availability and CASA model suitability for regional NPP estimation, the CASA model was selected to simulate the NPP of Guanzhong Basin and Hanzhong Basin from 1995 to 2014.

An eco-remote sensing coupling model, which has not yet been developed, could estimate the water retention of vegetation. At present, the most commonly used method to calculate WR is to divide the vegetation into different layers (the canopy

interception layer, litter layer holding water and soil layer), and the total WR is the sum of the three layers. In this study, the integrated storage capacity method was selected based in the actual situation and special location of the study area.

The most widely used revised universal soil equation (RUSLE) was used to calculate the SC, RUSLE was established from the soil erosion theory and observational data of natural runoff in the case of gentle slopes in small study areas. Therefore, it is consistent with the measured data from China only in cases where slopes are less than 14 °, the difference is very large when

a slope is greater than 14 °. If the SC model was used in area with a greater slope, its reliability would inevitably decrease; therefore, we needed to revise the formula to achieve values closer to the actual values. Based on the location and landforms in the study area, we used the method of Liu et al. (2000) to revise the terrain factor in the RUSLE model(Li et al., 2015; Liu et al., 2000). We thus proposed that the slope length factor index be revised to 0.5, which corrects for the L and S factors.

**Table 2**

Methods and calculation processes for assessing ES

| ES | Method | Calculation |
|---|---|---|
| FP | LRGO & NDVI (Kuri et al., 2014; Peng et al., 2017) | Grain $=a \times DN + b$ <br> Grain= food production <br> DN= value of NDVI |
| NPP | CASA model (Potter et al., 1993; Wang et al., 2017; Zhu et al., 2006) | $NPP(x, t) = APAR(x, t) \times \varepsilon(x, t)$ <br> $NPP(x, t)=$ net primary production of location x at month t, $APAR(x, t)=$ canopy absorbed incident solar radiation (MJ·m$^{-2}$) <br> $\varepsilon(x, t)=$ light utilization efficiency (g·C·MJ$^{-1}$). |



| | | |
|---|---|---|
| WR | ISCM<br>(Wang et al., 2017) | WR = $Q_1$ + $Q_2$ + $Q_3$<br>WR= amount of soil water retention<br>$Q_1$=canopy interception<br>$Q_2$=litter containment<br>$Q_3$=soil containment<br>$\mu$ =an equivalent value of rain, which relates to the amount of rainfall and the times of rain in a rain season. |
| SC | RUSLE model<br>(Li et al., 2015; Liu et al., 2000;<br>Wischmeier and Smith, 1978) | SC = f R K L S (1-C P)<br>SC= amount of soil conservation (t hm$^{-2}$ year$^{-1}$);<br>f=conversion coefficients;<br>R=rainfall erosivity factor (MJ mm hm$^{-2}$ h$^{-1}$ year$^{-1}$);<br>K=soil erodibility factor (t ha h ha$^{-1}$ MJ$^{-1}$ mm$^{-1}$);<br>L= slope length factor;<br>C=dimensionless crop and management factor;<br>P= conservation practice factor. |

Note: LRGO & NDVI: Linear relationship between grain output and NDVI; WR: Water retention; NPP: Net primary production; FP: Food production; SC: Soil conservation; NDVI: Normalized Difference Vegetation Index. CASA: Carnegie-Ames-Stanford Approach; ISCM: Integrated storage capacity method; RUSLE: Revised universal soil loss equation.

**2.4.Spatial correlation Analysis**

5   The paper use Spearman's coefficient to quantify the relationships among ES (Li et al., 2017; Su et al., 2012).Since the Spearman correlation coefficient does not require the distribution of the original variables, it is a non parametric statistical method, and the scope of application is wide, its expression is:

$$r_{xy} = \frac{\sum_{i=1}^{n}\sum_{j=1}^{n}\left(x_{ij}-\overline{x}\right)\left(y_{ij}-\overline{y}\right)}{\sqrt{\sum_{i=1}^{n}\sum_{j=1}^{n}\left(x_{ij}-\overline{x}\right)^2}\sqrt{\sum_{i=1}^{n}\sum_{j=1}^{n}\left(y_{ij}-\overline{y}\right)^2}} \qquad (1)$$

where $r_{xy}$ is the spatial correlation coefficient, with values ranging from -1 to 1. $r_{xy} > 0$ is a positive correlation, and indicates
10   that the two services are synergetic; $r_{xy} < 0$ is a negative correlation, which means that there are trade-offs between the two services. $x_{ij}$ and $y_{ij}$ represent the grid values in different ES spatial data.



## 3. Results

### 3.1. Correlation of FP, WR, NPP and SC

Taking Guanzhong Basin and Hanzhong Basin as the research objects, the average value of four types of services and the correlation coefficients among the four services were calculated from 1995-2014 using the ArcGIS 10.1 ->Spatial Analyst

Tools ->Multivariate ->Band Collection Statistics calculation software (Table 3). As shown in Table 3, both Guanzhong Basin and Hanzhong basin had ES correlations: there was a significant negative correlation between FP and NPP, FP and WR, and FP and SC, which is a trade-off. There was a positive correlation between NPP and WR, as well as between WR and SC, which is a synergism. The trade-off relationships between FP and NPP in the two basins were the most significant.

Comparing the relationships between trade-offs among the ES of two basins, it was found that the trade-off relationship

between FP and NPP in Guanzhong Basin was the most significant; the correlation coefficients were -0.40 and -0.31, respectively, and the trade-offs in Guanzhong Basin were stronger than those of Hanzhong Basin. The trade-off relationships between FP and WR, as well as between FP and SC in Guanzhong Basin were weaker than those in Hanzhong Basin. Comparing the relationship between the synergies of the two basins in ES, the synergistic relationships between NPP and WR, as well as between WR and SC in Hanzhong Basin were stronger than those in Guanzhong Basin; however, the synergistic

relationship between NPP and SC was weaker than that of Guanzhong Basin.

**Table 3**

The correlation coefficients of the annual mean values of ES in two Northwestern basins

|  | Guanzhong Basin | | | |  | Hanzhong Basin | | | |
|---|---|---|---|---|---|---|---|---|---|
|  | FP | NPP | WR | SC |  | FP | NPP | WR | SC |
| FP | 1 | -0.40** | -0.05** | -0.22** | FP | 1 | -0.31** | -0.22** | -0.31** |
| NPP |  | 1 | 0.09** | 0.15** | NPP |  | 1 | 0.29** | 0.06** |
| WR |  |  | 1 | 0.21** | WR |  |  | 1 | 0.27** |
| SC |  |  |  | 1 | SC |  |  |  | 1 |

Note: The significance tests were implemented using SPSS 23.0. **R** is the correlation coefficient. **significant at $p < 0.01$.

### 3.2. Time trade-offs or synergies for paired ES

To quantitatively analyse the time trade-offs and synergies among FP, NPP, WR and SC, with Guanzhong Basin and Hanzhong Basin as the research objects, and taking 1995-2014 as a time series, four types of ES were selected for paired ES analysis, and six groups of ES-related values were obtained (Fig. 5). The correlation coefficients shown in the figure were checked according to significance. The change of the correlation coefficient was used to judge the change of trade-offs and synergies. During 1995-2014, the average correlation coefficient of WR and FP in Guanzhong Basin was -0.10 with a standard deviation

of 0.18 and the average correlation coefficient of WR and FP in Hanzhong Basin was -0.25 with a standard deviation of 0.11.



The trade-offs between WR and FP in the two basins were significant, and the trade-off relationship of Hanzhong Basin was more significant than that of Guanzhong Basin, however, the trade-off relationship in Hanzhong Basin was more stable than that of Guanzhong Basin. The trade-off relationships between WR and FP in Guanzhong Basin and Hanzhong Basin showed fluctuating decreasing trends of 0.014 / a and 0.006 / a, respectively (Fig. 5 WR vs. FP).

The average correlation coefficient of WR and NPP in Guanzhong Basin was 0.23, with a standard deviation of 0.17. The average correlation coefficient of WR and FP in Hanzhong Basin was 0.35 with a standard deviation of 0.17. The synergistic relationship of Hanzhong Basin was more significant than that of Guanzhong Basin, but the synergistic relationship in Hanzhong Basin had the same stability as that of Guanzhong Basin. The synergistic relationships between WR and FP in Guanzhong Basin and Hanzhong Basin showed decreasing trends of 0.013/a and 0.009/a, respectively (Fig. 5 WR vs. NPP).

The average correlation coefficient of WR and SC in Guanzhong Basin was 0.35, with a standard deviation of 0.16, and the average correlation coefficient of WR and FP in Hanzhong Basin was 0.38, with a standard deviation of 0.14. WR and FP in both basins were strongly correlated. The synergistic relationships of Guanzhong Basin were more significant than the Guanzhong Basin and the synergistic relationships of Guanzhong Basin was more durable than that of Hanzhong Basin. The synergistic relationships between water retention and food supply in Guanzhong Basin and Hanzhong Basin showed

fluctuating decreasing trends of 0.011/a and 0.005/a, respectively (Fig. 5 WR vs. SC).

The average correlation coefficient of FP and NPP in Guanzhong Basin was -0.33, with a standard deviation of 0.04, and the average correlation coefficient of WR and FP in Hanzhong Basin was -0.20 with a standard deviation of 0.14.        The correlation was strong, the trade-offs between WR and FP in the two basins were significant and that of Guanzhong Basin was more significant than that of Hanzhong Basin, and the trade-off relationships between the two basins were significant. The

trade-off relationship in Guanzhong Basin was more stable than that of Hanzhong Basin. The trade-off relationship between FP and NPP in Guanzhong Basin and Hanzhong Basin showed decreasing trends of 0.001/a and 0.011/a, respectively (Fig. 5 FP vs. NPP).

The average correlation coefficient of FP and SC in Guanzhong Basin was -0.29 with a standard deviation of 0.49, and the average correlation coefficient of FP and SC was -0.29, with a standard deviation of 0.60. The correlation was strong when

the trade-offs between FP and SC in the two basins were significant; Guanzhong Basin and Hanzhong Basin had the same significance, and the trade-offs relationship in Guanzhong Basin was more stable than that of in Hanzhong Basin. The trade-off relationship between FP and SC in Guanzhong Basin showed a decreasing trend of 0.002/a, and the trade-off relationships between FP and SC in Hanzhong Basin showed fluctuating decreasing trends of 0.011/a and 0.005/a, respectively (Fig. 5 FP vs. SC).

The average correlation coefficient of NPP and SC in Guanzhong Basin was 0.35 with a standard deviation of 0.01, and the average correlation coefficient of FP and SC in Hanzhong Basin was 0.14 with a standard deviation was 0.18. Guanzhong Basin was more significant than Hanzhong Basin. The synergistic relationship of Guanzhong Basin was more durable than that of Hanzhong Basin. The synergistic relationship between NPP and SC in Guanzhong Basin showed a decreasing trend of



0.006/A, and the synergistic relationship between NPP and SC in Hanzhong Basin showed a decreasing tendency of 0.004/a (Fig. 5 NPP vs. SC).

In summary, the trade-offs between WR and FP and synergetic relationships between WR and NPP, as well as between WR and SC gradually weakened in the two basins, and the trade-offs and synergic relationship of Hanzhong Basin were more significant than those of Guanzhong Basin. The relationships between FP and SC, and between NPP and SC had a gradually decreasing trend in Guanzhong Basin, and a gradually increasing trend in Hanzhong Basin. The trade-off relationship between FP and SC in Guanzhong Basin was more significant than that in Hanzhong Basin, and the trade-off relationship between NPP and SC was the same expressive in two basins. The trade-off relationship between FP and NPP showed a small gradual tendency in the two basins, and the trade-off relationship of Guanzhong Basin was more momentous than that of Hanzhong Basin.

### 3.3. Spatial trade-offs or synergies for paired ES.

Using the powerful numerical calculation function of Matlab2014a software, spatial analysis and the powerful functions of ArcGIS 10.1 software, this paper analysed the spatial trade-offs and synergies for pairwise ES interactions in Guanzhong Basin and Hanzhong Basin by analysing the FP, NPP, WR and SC services from 1995 to 2014. Then, the grid was assigned calculated coefficients, exported as a TIFF file, cliped and drawn with ArcGIS 10.1 software (Fig. 6).

For FP and SC, strong trade-offs were spatially aggregated in the south-wester of Boji county, Taibai county, Fengxiang county, and Zhouzhi county, which accounted for 25.6 % of the LUCC (Fig. 6a). The spatial patterns of the interactions between FP and NPP were similar to the FP and SC interactions (Fig. 6e). The most pronounced distinction was that, the strong synergies between WR and NPP accounted for 5.8 % of the LUCC in the central and eastern region of Guanzhong Basin and in the middle and south of Hanzhong Basin (Fig. 6c). The spatial patterns of the interactions between SC and NPP were similar to the WR and NPP interaction (Fig. 5f). The spatial synergies and trade-offs between FP and WR, as well as between WR and SC was widely existed in two basins (Fig. 6b; Fig. 6d). The relationship between FP and WR in the two basins were mainly concentrated in the middle of cultivated land, urban impervious areas, and other types areas. The spatial synergies between WR and SC were mainly concentrated in the areas of farmland and urban impervious areas in the middle of the two basins.

Overall, the synergies among the four ES were mostly in the southwest or northwest of the two basins (e.g., woodland and grassland), whereas trade-offs mostly occurred in the middle or east of the two basins (e.g., farmland and built-up land).

## 4. Discussion

### 4.1. Combination of spatial analysis and spatial statistics

The relationships among ES were complex (Wang et al., 2017) and were mainly for trade-offs, synergies and no relationships. Among them, the current methods of verifying the relationships among ES mainly consist four types: methods of statistical analysis (e.g., correlation analysis, regression analysis, cluster analysis, redundancy analysis), spatial analysis (e.g., geographic





information system (GIS) technology), scenario simulation analysis (e.g., different land use dynamics, agricultural management, forest management measures and other scenarios of ES under dynamic simulation) and service liquidity analysis (e.g., network analysis technology). In this study, we used GIS and correlation analysis to verify the relationships among FP, NPP,

WR and SC. First, because of the wide range of data sources of the statistical methods, socio-economic statistics and survey data, geological environmental monitoring data, and assessment data, can be used for trade-offs and synergies among ES(Bai et al., 2011; Su et al., 2012). then, correlation analysis of statistics is a very useful way to measure the correlation and strength between two variables. The trade-offs and synergies among ES can be judged by the value of the correlation coefficients(Li et al., 2017).

**4.2. Comparative analysis and explanation of ES interactions at fine spatial and temporal scales**

Our study found that the trade-off between FP and NPP were the most significant in the two basins and that the trade-off relationship for Guanzhong Basin (R = -0.4, P, <0.01) was stronger than that for the Hanzhong Basin (R=-0.31, P<0.01). The main features of Guanzhong Basin were: small terrain slopes and a well-developed economy. Guanzhong Basin is mainly farmland and its farmland area was reduced, while its FP was increased, which was the result of improving the production

capacity of farmland. However, Hanzhong Basin is dominated by woodland and grassland (Fig. 3) and because of the climate change and variation of land use intensity in the last 20 years(Long, 2014; Plain et al., 2015), FP has rapidly increased, and woodland has been seriously damaged because of forests damages from the reclamation of land and project construction (Li et al., 2017), which led to the rapid decline of NPP. Therefore, the trade-offs between FP and NPP in the two basins are were particularly significant. In 2002, China implemented the Green for Grain Project and some of the sloping farmland was

converted into woodland and grasslands, which made the woodland and grassland area of Hanzhong Basin obviously increase. The increase of woodland and grassland areas was beneficial to the increase of NPP, WR and SC, which was also the main reason that the trade-offs between FP and NPP in Hanzhong Basin were less than those in Guanzhong Basin.

The proportion of woodland and grassland in Hanzhong Basin is larger than that in Guanzhong Basin; therefore, destruction by forest reclamation, engineering construction and other serious damage was more serious. WR and SC were significantly

reduced. However, NPP, WR, and SC were relatively small because of the small proportion of the woodland area in Guanzhong Basin. In summary, the trade-offs between FP and WR, as well as between FP and SC in Guanzhong Basin were weaker than those in Hanzhong Basin. The synergies between NPP and WR, as well as between WR and SC in Guanzhong Basin were weaker than those in Hanzhong Basin (see Table 1). However, this study found a different phenomenon; FP in two basins increased in the past 20 years, whereas SC, WR and NPP did not decline. The appearance of this phenomenon is not the rule

of negative trade-offs and synergies, but rather the consequence of the occurrence under certain preconditions. The two basins were in the stage of vegetation restoration during 2001-2014(Li et al., 2017; Tian et al., 2016; Yang et al., 2016). "The land area and capacity to provide NPP " exceededs the "land area to provide services", which may have been the main cause.



We also found that the synergistic relationship between NPP and SC showed a decreasing trend in Guanzhong Basin and an increasing in Hanzhoung Basin. This agrees with previous studies(Li and Wang, 2018; Li et al., 2017; Su et al., 2012; Sun and Li, 2017; Tian et al., 2016), and the differences of the synergistic relationships between NPP and SC were mainly because of the SC model, which is a function of vegetation and slope. Because of the small terrain slope, the SC capacity of Guanzhong Basin is small, but the slope and the vegetation coverage of Hanzhoung Basin are large and the SC capacity is also large. In the Green for Grain Project, the increase in woodland and grassland was more sensitive to the SC in Hanzhong Basin and less sensitive to that in Guanzhong Basin. Because of the increase of grassland in Hanzhong Basin, the NPP and the SC are obviously increased, and the synergistic relationship between NPP and SC thus showed an increasing trend. The NPP and SC increased in Guanzhong Basin because of the increase of woodland and grassland. Compared with the increasing speed of NPP, the SC decreased and the synergistic relationship between NPP and SC in Guanzhong Basin appears to have gradually decreased.

Our results show that the synergies and trade-offs between ES were widespread in the two basins. For example, the trade-off relationships between FP and WR in the two basins were mostly concentrated in the middle farmland and urban impervious areas, whereas the other areas had a synergistic relationship. Another example is that the spatial synergetic relationship between WR and SC was mainly concentrated in the farmland area of the middle of the two basins as well as urban impervious areas, whereas the other areas had the trade-offs relationships. The main reason here for these relationships was that scale plays an important role in quantifying ES and their interactions. For example, statistical methods, time trade-offs, and time synergy only showed a significant trade-off or synergy among the four ES in our study. However, spatial trade-offs and synergies revealed the relationships among the four ES in different locations or for different land use types, mainly because the statistical method of calculating ES interactions often does not take into account the temporal and spatial variabilities of local drivers, such as political policy, socio-economic factors and geographical spatial patterns (Costanza et al., 2014; Delphin et al., 2016; Fioramonti, 2017; Wang et al., 2017). Therefore, the interaction of ES at microscopic scales may be hidden on the macroscopic scale.

Our analysis revealed the trade-offs among ES in built-up land and farmland. Mainly because the region is located in the centre of Guanzhong Basin, farmland and grassland were the main LUCCs, and because of the increase of the population, the expansion of the urban area and deforestation led to the strong trade-offs between the supply service and regulation service. Surprisingly, we found that the synergies among ES rarely occurred in woodland areas, mainly because of the widespread reforestation in sub-humid areas, which may have led to accelerating evaporation, followed by water shortages in afforestation areas and potentially conflicting demands for water between ecosystems and humans. This result prompted us to rethink the rationality of restoration planning in sub-humid areas (Hamel and Bryant, 2017; Li and Wang, 2018; Tian et al., 2016; Wendland et al., 2010; Yang et al., 2016).



### 4.3. Limitations and future research directions

When first evaluating ES, many models were used in this study, including RUSLE, the water balance equation, the photosynthesis equation, and the spatialization of grain output based on NDVI. In these techniques, some uncertainties may exist regarding data collection or index selection. Unfortunately, there currently no effective solutions to these problems and these methods are still widely applied (Li and Zhou, 2016). For example, in the RUSLE model, the LS factor is generally derived from DEM data; however, the RUSLE model was established using natural runoff plot observations in the case of gentle slopes (Li et al., 2015; Liu et al., 2000). The slope length calculated by the formula coincides with the measured data in China only if the slope is less than 14 °; the slope length calculated by the formula is different from the measured data in China when the slope is greater than 14 °. If the calculation method of gentle slope is still adopted (the slope is less than 14 °), the credibility of the formula must be reduced, and we thus needed to revise the formula to approach the actual slope length. Therefore, the LS factor adopted in this study was from Liu et al. (2000), and the RUSLE model of the LS factor was modified. Comparing the results before and after LS correction, Wang et al. (2007) proved that the modified method has high accuracy (Wang et al., 2007), which provided the theoretical basis for selecting the LS calculation method in this experiment. However, because of the difference of the topographic conditions in the different regions, the relationship between the slope length and erosion has regional heterogeneity and the relative deviation of the model thus remained large; development of a suitable model needs further improvement and research. Another limitation of this study was the explanation of the driving mechanism of ES action. Although we determined the spatial pattern of ES interaction, it was difficult to clearly understand the causal relationship between ES and its driving factors, such as LULC change, climate change and regional hydrological status. This information is important for targeted engineering or protective measures to avoid trade-offs in ES.

Another limitation of this paper was in the comparison and analysis of the spatio-temporal trade-offs and synergies between the two basins for FP, NPP, WR, and SC. It is necessary to add more ES for comparative analysis in the future, which will help to clarify the services and their influence mechanisms. Therefore, it is necessary to study ES in the future considering the impact mechanisms and future scenario predictions (Cervelli et al., 2017; Kubiszewski et al., 2017; Peng et al., 2017; Sun and Li, 2017; Yang et al., 2016). For example, the study of LUCC and land allocation aims to determine how to affect the supply capacity of ES and, via stimulation of ES in different scenarios, to seek a win-win situation for the government's ecological planning and construction providing recommendations. The existing ecological protection policy, increased NPP, WR, SC and other regulatory functions but also led to the decline of FP and other supply services. These changes will lead to some ES that cannot meet the basic needs of human beings, which requires the development of society and human health by seeking a rational allocation of resources, to ensure the safety of ecological environments and achieve a balance between ecological and social development. For example, Guanzhong Basin has rapidly developed, in recent years and is thus faced with the dual pressures of population growth and urban expansion, an increasing demand for resources, more complicated ecological problems, and a direct threat to regional food security. It is very important to set an ecological protection red line, construct an



ecological security pattern, curb the trend of ecological environmental degradation, and promote the balanced development of population, resources and environment.

The current research on ES is mainly based on the theories and methods of geography and ecology, and research on quantitative evaluation of the dynamic relationship between ES using the theory and method of economics and management is obviously

insufficient(Dai et al., 2016). Therefore, future research on synergies and trade-offs need to better use the theory and methods of economics and management to integrate ecosystem service "geography" and ecosystem service "social economy" research.

## 5. Conclusions

Our study compared the fine-scale spatio-temporal relationship among four ESs (FP, WR, NPP, and SC) from 1995 to 2014 in Guanzhong Basin and Hanzhong Basin in Shaanxi Province, China. The relationships were generally of three

types: trade-offs, synergies and no relationship. Our results demonstrated that the synergies and trade-offs among ES were prevalent in the two basins rather than purely trade-offs or synergies. We also found that the synergies between ES rarely occurred in woodlands, a result that is inconsistent with the results of previous studies, which prompted us to develop the problems of ES and rethink the rationality of local vegetation restoration planning. Based on a comparative study of the two basins, this research analysed the difference between an economically developed region (Guanzhong

Basin) and regional environmental protection region (Hanzhong Basin), which can provide a basis for the development of corresponding ES management policies in different regions.

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



**Figure 1: The research ideas and framework of this paper. FP: food production; NPP: net primary production; SC: soil conservation; WR: water retention; GIS: geographic information system; CASA: Carnegie-Ames-Stanford Approach; RUSLE: revised universal soil loss equation; DEM: digital elevation model; NDVI: normalized difference**

10   **vegetation index.**



**Figure 2: Locations of Guanzhong Basin and Hanzhong Basin and the study area. a:Guanzhong Basin and the administrative boundary of Guanzhong Basin; b:the location of Shaanxi in China; c: the location of Guanzhong Basin and Hanzhong Basin in Shaanxi Province and the two basins located on both sides of the dividing line between north and south; d: Hanzhong Basin and its the administrative boundary.**





**Figure 3: Temporal distribution of Landsat remote sensing images in Guanzhong Basin and Hanzhong Basin. The vertical axis is the month (above is the Landsat image month of Guanzhong Basin, which was the Landsat image month of the Hanzhong Basin), and the horizontal axis is the year (1995-2014).**





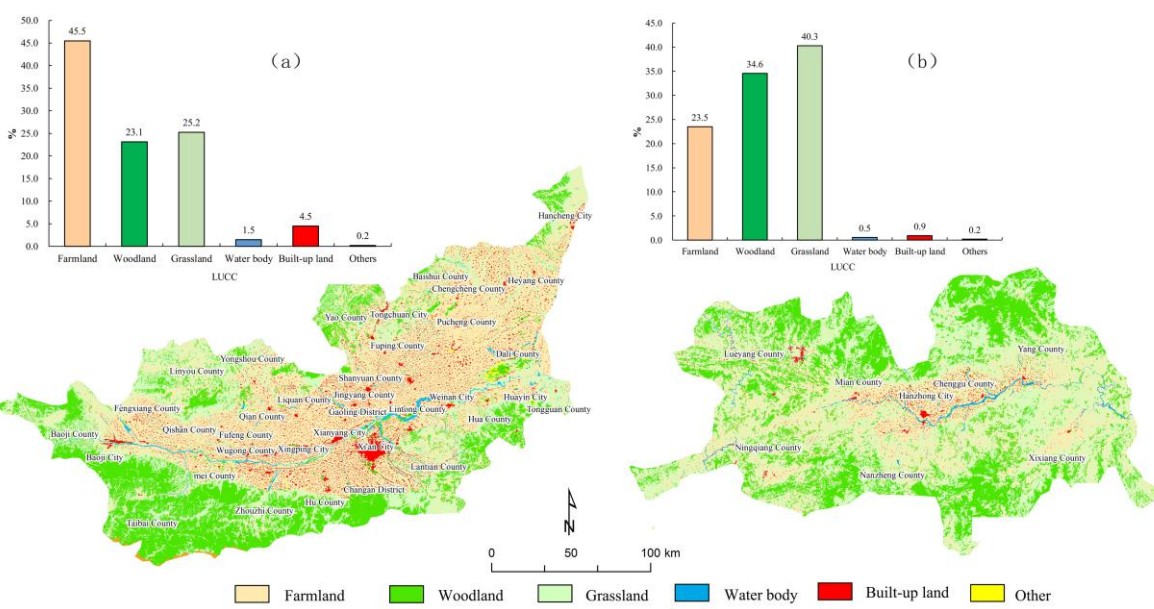

**Figure 4: LUCC types and the proportion of LUCC types. There were six LUCC types: farmland, water body, woodland, grassland, built-up land and other types. a: LUCC types and the proportion of the six LUCC types in Guanzhong Basin in 1995; b: LUCC types and the proportion of six LUCC types in Hanzhong Basin in 1995.**

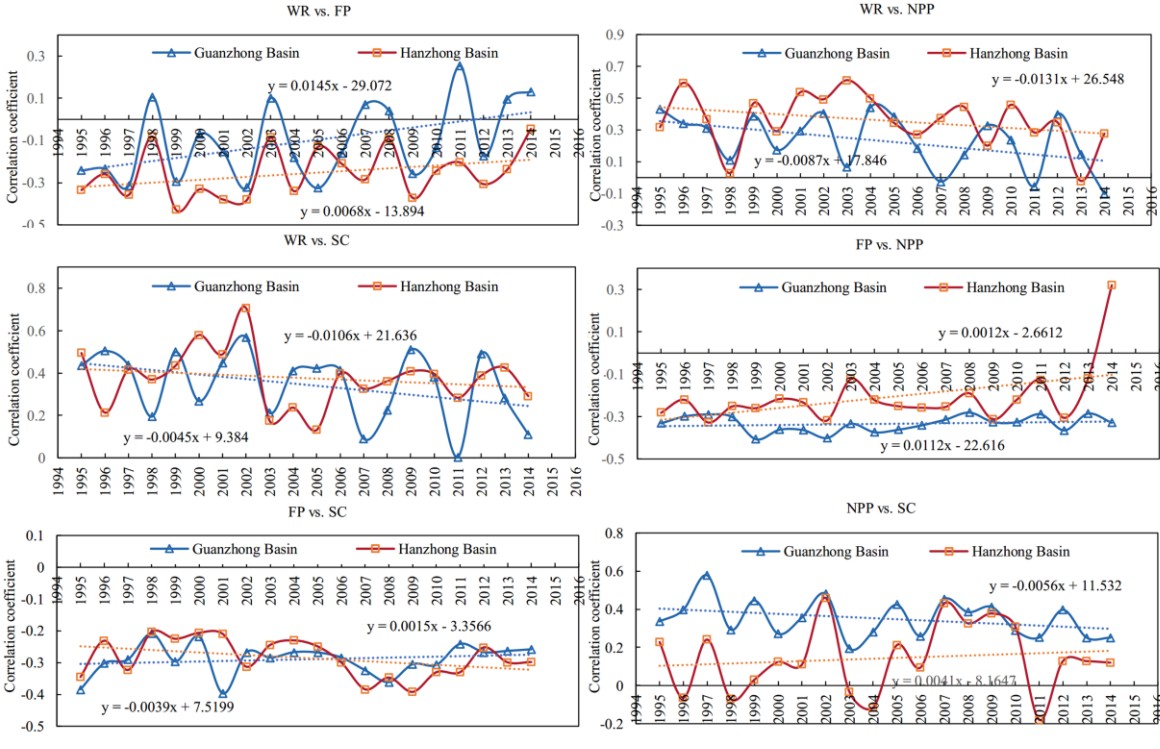

**Figure 5: Comparison of the time trade-offs and pairwise synergetic ES interactions in Guanzhong Basin and Hanzhong Basin from1995-2014. The quantitative relationships between pairwise ES was based on correlation analysis using**




**MATLAB 2014a software. Correlation analyses were all significant at the 0.01 level. WR: water retention; NPP: Net primary production; FP: Food production; SC: Soil conservation.**





(a) FP vs. SC    (b) FP vs. WR    (c) WR vs. NPP

Guanzhong Basin    Guanzhong Basin    Guanzhong Basin

Hanzhong Basin    Hanzhong Basin    Hanzhong Basin

(d) WR vs. SC    (e) FP vs. NPP    (f) SC vs. NPP

Guanzhong Basin    Guanzhong Basin    Guanzhong Basin

Hanzhong Basin    Hanzhong Basin    Hanzhong Basin

Trade-off **    Trade-off *    Trade-off    No relationship

Synergy **    Synergy *    Synergy    0    100    200 km



**Figure 6: Comparison of the spatial trade-offs and synergies for paired ES in Guanzhong Basin and Hanzhong Basin.** ∗∗**Correlation were all significant at the 0.01 level;** ∗**Correlation were all significant at the 0.05 level; other correlations were weak or not significant. WR: water retention; NPP: Net primary production; FP: Food production; SC: Soil conservation.**