# Peer review of "Comparative analysis of trade-offs and synergies in ecosystem services between Guanzhong Basin and Hanzhong Basin in China"

_Biogeosciences, 2018_

## Referee Comment (RC1) · Anonymous Referee #2 · 20 Mar 2018

This study examined the spatial and temporal relationship among four main ecosystem services (ES), including food production (FP), net primary production (NPP), water retention (WR) and soil conservation (SC), of two basins in central china. This study found a trade-off relationships between FP and NPP, and synergistic relationships between NPP and WR, as well as between WR and SC. I think the methods are generally sound, and support the results. The manuscript is easy to follow, and also fit the scope of the journal.

I have several major concerns as follows: 1. This analysis relied heavily on models to quantify different ES, and therefore each model will have its own uncertainty.

The authors have acknowledged this in the manuscript. However, a formal uncertainly analysis for each model and how uncertainty of each model will propagate to the main results will help reader understand the results better. 2. The trade-off (negative) relationship between FP and NPP is a little bit hard to understand since both FP and NPP is a function of NDVI. Some more explanation on this will help reader to understand the result better. 3. P7L10: $r_{xy}$ïij$\check{d}$0 indicates positive (synergetic) relationship, and $r_{xy}$< 0 indicates negaitve (tradeoff) relationship. Do you have a significant level here? For example, $r_{xy}$ïij$\check{d}$0 and $p < 0.01$ indicates positive (synergetic) relationship, and $r_{xy}$< 0 and $p < 0.01$ indicates negative (tradeoff) relationship. While $p > 0.01$ is no relationship. 4. This manuscript need a language edits by native speaker.

Some other observations: 1. P7 table: how WR was calculated for different layers? 2. P8 line 3-5, P8 line 20-22, P8 line 11-15: these sentences belong to method section 3. Figure 3 can be in an appendix.

---

## Author Comment (AC1) · 12 Apr 2018

We thank the reviewer for the constructive comments. The following are our point by point response to these comments.

Major issues:

1. This analysis relied heavily on models to quantify different ES, and therefore each model will have its own uncertainty. The authors have acknowledged this in the manuscript. However, a formal uncertainly analysis for each model and how uncertainty of each model will propagate to the main results will help reader understand the

results better.

Author's response: Thank you very much for your comment. In this study, many models were used to evaluate ecosystem services, including LRGO & NDVI, RUSLE, and CASA. In these models there might exist some uncertainties about data collection or index selection. Unfortunately, there are currently no effective ways to solve these problems, and these methods are still widely used (Li and Zhou, 2016). The present study uses CASA model to simulate NPP. When selecting the data, error in land use interpretation, resolution ratio of remote sensing image, different precisions of auxiliary data and cut of pixel by vector data can cause data uncertainty.

Li, J. and Zhou, Z. X.: Natural and human impacts on ecosystem services in Guanzhong - Tianshui economic region of China, 6803–6815, https://doi.org/ 10.1007/s11356-015-5867-7, 2016.

2. The trade-off (negative) relationship between FP and NPP is a little bit hard to understand since both FP and NPP is a function of NDVI. Some more explanation on this will help reader to understand the result better.

Author's response: Thank you very much for your comment. Many models were used in this study, including NPP (CASA model) and FP (LRGO & NDVI model) based on NDVI. Unfortunately, there are currently no effective ways to solve these problems, and these methods are still widely used (Peng et al., 2017). Peng, J., Tian, L. and Liu, Y., et al.: Ecosystem services response to urbanization in metropolitan areas: Thresholds identification, Science of The Total Environment, 608, 706–714, https://doi.org/ 10.1016/j.scitotenv.2017.06.218, 2017.

3. P7L10: rxyïijd0 indicates positive (synergetic) relationship, and rxy< 0 indicates negaitve (tradeoff) relationship. Do you have a significant level here? For example, rxyïijd0 and p < 0.01 indicates positive (synergetic) relationship, and rxy< 0 Âŕand p < 0.01 indicates negative (tradeoff) relationship. While p> 0.01 is no relationship.

Author's response: Thank you very much for your comment. We have significant levels in the Results section (See table 3 and Figure 5). Based on your suggestion, We've added a significant level here (\*\*Correlation were all significant at the 0.01 level; \*Correlation were all significant at the 0.05 level; other correlations were weak or not significant.")

4. This manuscript need a language edits by native speaker.

Author's response: We have had the manuscript polished by hiring a professional editing agency.

Some other observations:

1. P7 table: how WR was calculated for different layers?

Author's response: Thank you very much. At present, the main method of estimating forest water retention is to decompose the water retention effect of different forest function layers, followed by canopy interception, litter water holding and soil water storage. The total capacity is the sum of the above mentioned three parameters. This model of water retention was modified by vegetation coverage. The formula is calculated by the following (Wang et al., 2017): $Q\_i = Q\_if + Q\_il + Q\_is$ $Q\_if = \alpha\_Li \times \beta\_Ci$ $Q\_il = \varepsilon\_Ci \times \beta\_Ci$ $Q\_is = \theta\_i \times \varphi\_i$ where $Q\_i$ is the ith grid of the water retention (mm), $Q\_if$ is the ith grid of the canopy interception (mm), $Q\_il$ is the ith grid of the litter water holding (mm), $Q\_is$ is the ith grid of the soil water storage (mm), $\alpha\_Li$ is the ith grid of the maximum intercept in the canopy and understory shrubs times of the process of precipitation (mm), $\beta\_Ci$ is ith grid of the forest coverage (%), $\varepsilon\_Ci$ is the ith grid of the litter grid maximum water holding capacity (mm), $\theta\_i$ is the ith grid of the soil non capillary porosity (%), and $\varphi\_i$ is the ith grid of the thickness of the soil (mm).

2. P8 line 3-5, P8 line 20-22, P8 line 11-15: these sentences belong to method section

Author's response: Thank you very much. We've moved these sentences (e.g. P8 line 3-5, P8 line 20-22, P8 line 11-15) to the method section

3.Figure 3 can be in an appendix. Author's response: Thank you very much. We have moved Figure 3 into appendix A.

Please also note the supplement to this comment:
https://www.biogeosciences-discuss.net/bg-2018-33/bg-2018-33-AC1-supplement.pdf

**Supplement:**

**Comparative analysis of trade-offs and synergies in ecosystem services between Guanzhong Basin and Hanzhong Basin in China**

Bo-Yan Li, Wei Wang, Yun-Chen Wang

[Figure]

**Figure A1: Temporal distribution of Landsat remote sensing images in Guanzhong Basin and Hanzhong Basin. The vertical axis is the month (above is the Landsat image month of Guanzhong Basin, which was the Landsat image month of the Hanzhong Basin), and the horizontal axis is the year (1995-2014).**

---

## Referee Comment (RC2) · Anonymous Referee #3 · 30 May 2018

Li and others study synergies and trade-offs amongst ecosystem services in two different watersheds in China. The justification of the choice of the two watersheds should be improved in the Introduction do describe how studying contrasting basins can benefit the theory and practice of ecosystem service science. The Discussion is a huge missed opportunity and doesn't describe the major findings and how they may apply to ecosystem service science and practice. It needs to be entirely re-written. English usage is largely good but the text can be more efficient and powerful. For example, in the abstract, "An important feature of the relationships among ecosystem services (ES) is they have temporal and spatial patterns" could be "Ecosystem services (ES) and relationships among them have temporal and spatial patterns that need to be un-

derstood." The paper makes many good points, but needs considerable revision before it is acceptable. Minor comments follow.

P 1 L 15: A 'good ecological environment' could mean different things to different people.

The sentence beginning P 1 L 13 is 6 lines long. And why choose these approaches? Also, there are far more than 4 types of ecosystem services. See Costanza 1997 for a primer.

P 1 line 30 is far too wordy.

P 2 L 1 what are 'life materials'?

P 2 L 3 start a new sentence.

P 2 L 6 what is a 'banned slope'?

Odd transition on P 2 L 12: no relationship between ES, which is how the previous sentence ends, need not be a dynamic change that threatens the world.

P 2 L 17 don't specify in a journal that a certain paper is published in Nature.

At the end of the Introduction, the justification for using the two study basins is poorly developed.

P 3 L 10: have not been given sufficient what? The arguments need to be presented more logically. The Introduction is mostly centered around what other studies haven't done to date rather than why the study of these two basins may be interesting and useful other than to state one has a 'good ecological environment', whatever that means.

'approximately 39064.5 square kilometres' is extremely specific! A number with six significant digits isn't approximate.

Figure 1 doesn't make sense. ES1…n is unnecessary, datasets can be died more clearly to the models, and the flow should be adjusted from top to bottom (or left to

right) highlighting the work flow.

Many readers are unlikely to know what "Chiangnan" refers to and why this is important. The fact that the basins are "famous" is immaterial. Many fonts in figure 2 are too small to be readable.

2.2: were there differences between Landsat 5, 7, and 8? Did the change from one to another induce artificial temporal patterns amongst variables? How were the decision trees implemented? I see later that it was eCognition. Describe this above when decision trees are first mentioned.

"A spatial resolution of 30 meters met the requirements of regional scale" doesn't make sense and at any rate would not be a statement for the methods section. Show the reader why, don't tell them.

Why are June and September the focus?

How did Google Earth provide verification and on what basis is it precise? Why not just use Google Earth if it's precise and verified?

It appears that the June and September deliniations perhaps have something to do with water bodies?

P 6 L 14: a few ecosystem models that interact with remote sensing (CLM if I'm not mistaken) include plant water capacity.

In Table 1, please use the multiplication sign instead of dots for the equation for SC.

P 6, "Carnegie", "Ames", and "Stanford" are all proper nouns and should be capitalized.

'The paper use' on page 7 L 5 is one of many examples of English usage that could benefit from improvement.

Table 2 and elsewhere: are the quantities (e.g. "FP") unitless? From the equations it appears that they should have units although it is correct that the corresponding

correlation coefficients wouldn't. A comparison amongst the basins of the quantities in Table 2 would be informative.

Avoid superlatives like 'powerful numerical calculation function' on page 10. Parts of section 3.3 is largely Methods and not Results as written.

Re-think this passage: The relationships among ES were complex (Wang et al., 2017) and were mainly for trade-offs, synergies and no relationships. It just means that relationships in all directions (positive, negative, and no relationship) was found.

'seriously damaged because of forests damages' needs to be re-worded.

The Discussion is mainly a discussion of limitations in the manuscript. Keep the important points about limitations, but also discuss what you found and what it means. In the conclusion for example this pops out, "We also found that the synergies between ES rarely occurred in woodlands, a result that is inconsistent with the results of previous studies." What other studies? The Discussion section is a huge missed opportunity to discuss what the study means in context and how it may improve theory and management of ecosystem services in these regions.

---

## Author Comment (AC2) · 2 Jun 2018

We thank the reviewer and editor for the constructive comments. The following are our point by point response to these comments.

Minor comments follow:

1. Li and others study synergies and trade-offs amongst ecosystem services in two different watersheds in China. The justification of the choice of the two watersheds should be improved in the Introduction do describe how studying contrasting basins can benefit the theory and practice of ecosystem service science. The Discussion is

a huge missed opportunity and doesn't describe the major findings and how they may apply to ecosystem service science and practice. It needs to be entirely re-written. English usage is largely good but the text can be more efficient and powerful. For example, in the abstract, "An important feature of the relationships among ecosystem services (ES) is they have temporal and spatial patterns" could be "Ecosystem services (ES) and relationships among them have temporal and spatial patterns that need to be understood." The paper makes many good points, but needs considerable revision before it is acceptable.

Author's response: Thank you very much. We paid special attention to above issue, and the relevant text is now used appropriately throughout the abstract and text of the revised manuscript. The statement "An important feature of the relationships among ecosystem services(ES) is they have temporal and spatial patterns" was corrected to read as "Ecosystem services (ES) and relationships among them have temporal and spatial patterns that need to be understood.". The corrections are shown in red (see P1 line11). We have rewritten the Discussion according to your comment (see p11 lines5-p13- line 7)

2. P 1 L 15: A 'good ecological environment' could mean different things to different people.

Author's response: Guanzhong Basin and Hanzhong Basin are both famous valley basins in northwest China. They are located on both sides of the dividing line between north and south (the Qinling-Huaihe River), respectively. A 'good ecological environment' could mean different regional ecosystems have different ES interactions (Xu et al. 2018). Thus, the effects of regional differences on ES interactions need research, to improve the horizontal comparisons between different regions and understand more about the links between multiple ES and the strength and direction of their interactions. Li, Z. peng, Long, Y. qiao, Tang, P. qin, Tan, J. yang, Li, Z. guo, Wu, W. bin, et al. (2017). Spatio-temporal changes in rice area at the northern limits of the rice cropping system in China from 1984 to 2013. Journal of Integrative

Agriculture, 16(2), 360–367. doi:10.1016/S2095-3119(16)61365-5 Lim, T. (2000). A Comparison of Prediction Accuracy , Complexity , and Training Time of Thirty-three Old and New Classification Algorithms. Machine Learning, 40, 203–229. Loh, W.-Y., & Shih, Y.-S. (1997). Split Selection Methods for Classification Trees. Statistica Sinica, 7(4), 815–840. doi:10.2307/24306157 Xu, X., Yang, G., Tan, Y., Liu, J., & Hu, H. (2018). Ecosystem services trade-offs and determinants in China's Yangtze River Economic Belt from 2000 to 2015. Science of the Total Environment, 634, 1601–1614. doi:10.1016/j.scitotenv.2018.04.046

3. The sentence beginning P 1 L 13 is 6 lines long. And why choose these approaches? Also, there are far more than 4 types of ecosystem services. See Costanza 1997 for a primer.

Author's response: The trade-offs or synergies among multiple ESs were more complex to calculate and understand. Using four ES indicators, FP, CS, WR and SC, as examples, the coupling relationships were extremely complicated to explore (Amador et al. 2005). Therefore, their trade-offs were determined by these complex processes among the four variables. Future efforts should focus on the multiple interactions (17 types of ecosystem services) as well as the underlying processes.

Amador, J. A., Görres, J. H., & Savin, M. C. (2005). Role of soil water content in the carbon and nitrogen dynamics of lumbricus terrestris, l. burrow soil. Applied Soil Ecology, 28(1), 15-22.

4. P 1 line 30 is far too wordy.

Author's response: Ecosystem services (ES) are the benefits people derive from ecological processes of a non-human nature, where all benefits are directly or indirectly derived from the ecosystems (Costanza et al., 1997; Daily, 1997; Millennium Ecosystem Assessment (MEA), 2005; The Economics of Ecosystems and Biodiversity (TEEB), 2010). The corrections are shown in red (see P1 line 30-P2 line 1).

5. P 2 L 1 what are 'life materials'?

Author's response: The statement "'the cycle of life materials" was corrected to read as "the material circulation". The corrections are shown in red (see P2 line 1).

6. P 2 L 3 start a new sentence.

Author's response: P 2 L 3 have started a new sentence: "Moreover, they are the resource and environment foundation for the existence and development of human society".

7. P 2 L 6 what is a 'banned slope'?

Author's response: The statement "'reclaiming banned slope" was corrected to read as "reclaiming slopes banned by the State shall be prohibited". The corrections are shown in red (see P2 line 6).

8. Odd transition on P 2 L 12: no relationship between ES, which is how the previous sentence ends, need not be a dynamic change that threatens the world.

Author's response: The statement "'no relationship between ES, which is how the previous sentence ends, need not be a dynamic change that threatens the world." was corrected to read as "No relationship means that changes in one do not affect the other". The corrections are shown in red (see P2 line 12).

9. P 2 L 17 don't specify in a journal that a certain paper is published in Nature. At the end of the Introduction, the justification for using the two study basins is poorly developed.

Author's response: We have deleted a journal name and correspondingly corrected the Introduction section (see P2 lines 13-14). Guanzhong Basin and Hanzhong Basin are both famous valley basins in northwest China. They are located on both sides of the dividing line between north and south (the Qinling-Huaihe River), respectively. This research was to study the spatial and temporal characteristics of the synergies

and trade-offs in ES in Guanzhong Basin, which has a good ecological environment (poorly developed), and the economically developed Guanzhong Basin, as well as to compare the ES differences between the two basins.

10. P 3 L 10: have not been given sufficient what? The arguments need to be presented more logically. The Introduction is mostly centered around what other studies haven't done to date rather than why the study of these two basins may be interesting and useful other than to state one has a 'good ecological environment', whatever that means.

Author's response: We have summarized the background and bibliography for a nearly 20-year period, and we have rewritten this part according to your comment (see P3 lines 6-13). "However, the trade-offs and synergies analysis still remain poorly understood due to three major challenges: (1) a historical approach mostly compare two snapshots in time, and rarely investigate interactions among multiple services through time (Renard et al., 2015); (2) the studies of trade-offs or synergies have mainly been based on quantitative analysis of statistical relations to reflect regional overall differences, and there has been a lack of spatial expression of temporal and spatial differences within the region; and (3) trade-offs and synergies have rarely been considered in the study of spatio-temporal contrastive analysis" were added.

11. approximately 39064.5 square kilometres' is extremely specific! A number with six significant digits isn't approximate.

Author's response: "approximately" was deleted (see P3 line 20).

12. Figure 1 doesn't make sense. ES1: : :n is unnecessary, datasets can be died more clearly to the models, and the flow should be adjusted from top to bottom (or left to right) highlighting the work flow.

Author's response: The flow has been adjusted from top to bottom.

13. 2.2: were there differences between Landsat 5, 7, and 8? Did the change from

one to another induce artificial temporal patterns amongst variables? How were the decision trees implemented? I see later that it was eCognition. Describe this above when decision trees are first mentioned

Author's response: There are a few differences between Landsat 5, 7, and 8. The three products are comparable with some initial considerations. As you can see (Table s1), the resolutions of the products are more or less the same. However, the combinations used to create RGB composites differ from Landsat 7 and Landsat 5. For instance, bands 4, 3, 2 are used to create a color infrared (CIR) image using Landsat 7 or Landsat 5. To create a CIR composite using Landsat 8 data, bands 5, 4, 3 are used. Table s1 Landsat-8 OLI and TIRS spectral bands compared to Landsat-5 TM and Landsat-7 ETM+ spectral bands.

The images were selected because the acquisition dates were close to the image dates. The remote sensing images was acquired and classified using the "Quick, Unbiased, Efficient Statistical Trees" (QUEST) decision tree algorithm. The algorithm is described and implemented in (Loh and Shih 1997) and the performance of this algorithm compared with other classification methods can be found in (Lim 2000) (see P4 lines 12-15)

14. "A spatial resolution of 30 meters met the requirements of regional scale" doesn't make sense and at any rate would not be a statement for the methods section. Show the reader why, don't tell them.

Author's response: "A spatial resolution of 30 meters met the requirements of regional scale" was deleted.

15. Why are June and September the focus?

Author's response: Because most of the remote sensing data were distributed in "June" or "September" (Appendix A, Fig. A1), and some data were not present or subject to cloudiness (e.g., year:1997; strip number:126; row number:035). "June" refers to a

similar month in June or around June, and the same was true for "September". Considering the seasonal variation of partial features and quality of the remote sensing data, the remote sensing data from the previous year's December or subsequent year's January data were replaced.

16. How did Google Earth provide verification and on what basis is it precise? Why not just use Google Earth if it's precise and verified?

Author's response: Google Earth high-resolution images, used for the discrimination and selection of auxiliary interpretation marks and an error check after classification (Li et al. 2017).

Li, Z. peng, Long, Y. qiao, Tang, P. qin, Tan, J. yang, Li, Z. guo, Wu, W. bin, et al. (2017). Spatio-temporal changes in rice area at the northern limits of the rice cropping system in China from 1984 to 2013. Journal of Integrative Agriculture, 16(2), 360–367. doi:10.1016/S2095-3119(16)61365-5

17. It appears that the June and September deliniations perhaps have something to do with water bodies?

Author's response: The area experiences a typically warm and semi-humid continental monsoon climate in which the four seasons are distinct and rainfall is moderate. Our main purpose in June and September was to eliminate the impact of seasonal changes on land use types. The June and September deliniations perhaps have something to do with water bodies. Therefore, water bodies were extracted by comparing the changes in June and September.

18. P 6 L 14: a few ecosystem models that interact with remote sensing (CLM if I'm not mistaken) include plant water capacity.

Author's response: The statement "An eco-remote sensing coupling model, which has not yet been developed, could estimate the water retention of vegetation." was corrected to read as "a few ecosystem models that interact with remote sensing (e.g.

Community Earth System Model (CESM)) include canopy hydrology". The corrections are shown in red (see P6 lines 18-19).

19. In Table 1, please use the multiplication sign instead of dots for the equation for SC.

Author's response: The statement "SC = fÂů RÂů KÂů LÂů SÂů (1-CÂů P)" was corrected to read as "SC = f × R × K × L×S×(1-C×P)". The corrections are shown in red (see Table 2).

20. P 6, "Carnegie", "Ames", and "Stanford" are all proper nouns and should be capitalized

Author's response: The statement "carnegie-ames-stanford approach (CASA) model" was corrected to read as "Carnegie-Ames-Stanford Approach (CASA) model". The corrections are shown in red (see P6 line 11).

21.'The paper use' on page 7 L 5 is one of many examples of English usage that could benefit from improvement.

Author's response: We have corrected the entire paper. Furthermore, we have had the manuscript polished by hiring a professional editing agency. The statement "The paper use Spearman's coefficient to quantify the relationships among ES" was corrected to read as "Spearman's rank correlation is employed to analyze the relationships between ES pairs at the grid-cells level". The corrections are shown in red (see P8 line 4).

22. Table 2 and elsewhere: are the quantities (e.g. "FP") unitless? From the equations it appears that they should have units although it is correct that the corresponding correlation coefficients wouldn't. A comparison amongst the basins of the quantities in Table 2 would be informative.

Author's response: Thank you very much. We have addressed this issue by carefully revising the Table 2 in the manuscript.

23. Avoid superlatives like 'powerful numerical calculation function' on page 10. Parts of section 3.3 is largely Methods and not Results as written

Author's response: "Using the powerful numerical calculation function of Matlab2014a software, spatial analysis and the powerful functions of ArcGIS 10.1 software, this paper analysed the spatial trade-offs and synergies for pairwise ES interactions in Guanzhong Basin and Hanzhong Basin by analysing the FP, NPP, WR and SC services from 1995 to 2014. Then, the grid was assigned calculated coefficients, exported as a TIFF file, cliped and drawn with ArcGIS 10.1 software (Fig. 6)" were deleted.

24. Re-think this passage: The relationships among ES were complex (Wang et al., 2017) and were mainly for trade-offs, synergies and no relationships. It just means that relationships in all directions (positive, negative, and no relationship) was found.

Author's response: Thank you very much. The statement "The relationships among ES were complex and were mainly for trade-offs, synergies and no relationships. It just means that relationships in all directions (positive, negative, and no relationship) was found." was corrected to read as "Relationships among ESs have been found to be complex and were classified trade-offs, synergies and without relationships". The corrections are shown in red (see P11 lines 10-11).

25. 'seriously damaged because of forests damages' needs to be re-worded

Author's response: Thank you very much. The statement "seriously damaged because of forests damages" was corrected to read as "seriously damaged by the reclamation of land and project construction". The corrections are shown in red (see P11 line 28).

26. The Discussion is mainly a discussion of limitations in the manuscript. Keep the important points about limitations, but also discuss what you found and what it means. In the conclusion for example this pops out, "We also found that the synergies between ES rarely occurred in woodlands, a result that is inconsistent with the results of previous studies." What other studies? The Discussion section is a huge missed

opportunity to discuss what the study means in context and how it may improve theory and management of ecosystem services in these regions.

Author's response: Thank you very much. "We also found that the synergies among ES rarely occurred in woodland areas, mainly because of the widespread reforestation in sub-humid areas, which may have led to accelerating evaporation, followed by water shortages in afforestation areas and potentially conflicting demands for water between ecosystems and humans. This result prompted us to rethink the rationality of restoration planning in sub-humid areas (Hamel and Bryant, 2017; Li and Wang, 2018; Tian et al., 2016; Wendland et al., 2010; Yang et al., 2016). Surprisingly, our analysis revealed the trade-offs among ES in built-up land and farmland. Mainly because the region is located in the centre of Guanzhong Basin, farmland and grassland were the main land use types, and because of the increase of the population, the expansion of the urban area and deforestation led to the strong trade-offs between the supply service and regulation service" were added in the Discussion section (see P13 lines 12-19) "We also found that the synergies between ES rarely occurred in woodlands, a result that is inconsistent with the results of previous studies." were deleted in Conclusions section.

We have rephrased and clarified the text per your suggestion.

Please also note the supplement to this comment:
https://www.biogeosciences-discuss.net/bg-2018-33/bg-2018-33-AC2-supplement.zip

| Datasets |
|---|
| National fundamental geographic information vector data \| Topographic data (e.g. DEM) |
| Meteorological data(e.g. Temperature, precipitation, solar radiation...) \| Soil properties |
| Statistical data \| Remote sensing image data (e.g. Landsat-5 TM, NDVI) \| Land use types |

**GIS-based regional model**

| Food production model | NPP model of natural vegetation(CASA model) |
|---|---|
| Modified RUSLE model | Water retention model |

**Ecosystem services**

| ES 1 → FP | ES 2 → CS ← NPP |
|---|---|
| ES 3 → SC | ES 4 → WR |

**ES interactions**

Trade-off    No relationship    Synergy

Temporal change    Spatial change

**Fig. 1.** The research ideas and framework of this paper.

**Fig. 2.** Locations of Guanzhong Basin and Hanzhong Basin and the study area.

[Figure]

**Fig. 3.** Land use types and the proportion of land use types.

[Figure]

**Fig. 4.** Comparison of the time trade-offs and pairwise synergetic ES interactions

**(a) FP vs. SC**

Guanzhong Basin

Hanzhong Basin

**(b) FP vs. WR**

Guanzhong Basin

Hanzhong Basin

**(c) WR vs. CS**

Guanzhong Basin

Hanzhong Basin

**(d) WR vs. SC**

Guanzhong Basin

Hanzhong Basin

**(e) FP vs. CS**

Guanzhong Basin

Hanzhong Basin

**(f) SC vs. CS**

Guanzhong Basin

Hanzhong Basin

Trade-off **     Trade-off *     Trade-off     No relationship

Synergy **     Synergy *     Synergy     0   100   200 km   N

**Fig. 5.** Comparison of the spatial trade-offs and synergies for paired ES in Guanzhong Basin and Hanzhong Basin.